# Efficient Distributed MLLM Training with Cornstarch

**Insu Jang**[1]  **Runyu Lu**[1]  **Nikhil Bansal**[1]  **Ang Chen**[1]  **Mosharaf Chowdhury**[1]

## Abstract

Multimodal large language models (MLLMs) extend the capabilities of large language models (LLMs) by combining heterogeneous model architectures to handle diverse modalities like images and audio. However, this inherent heterogeneity in MLLM model structure and data types makes makeshift extensions to existing LLM training frameworks unsuitable for efficient MLLM training, especially in distributed training.

In this paper, we present Cornstarch, an efficient distributed MLLM training framework that contemplates MLLM's unique characteristics in both model and data parallelization. Cornstarch introduces frozen-aware pipeline parallelism and workload-balanced context parallelism to improve MLLM training throughput. Our extensive evaluation shows that Cornstarch outperforms state-of-the-art solutions by $2.26\times$ on average in terms of MLLM training throughput.

Cornstarch is an open-source project available on Github [1].

## 1. Introduction

Multimodal large language models (MLLMs) aim to extend LLMs' reasoning capabilities to perform complex tasks across various data modalities, such as images and audio (Wang et al., 2024; Lin et al., 2024a;b; Chen et al., 2024b; Xu et al., 2021; Radford et al., 2023; Chu et al., 2024; AI, 2024; Chen et al., 2024a; Zhu et al., 2024; Zhang et al., 2024; Tong et al., 2024; Liu et al., 2023; 2024a; Abdin et al., 2024). While MLLMs can be trained from scratch like traditional LLMs, they are more commonly constructed by integrating *pretrained* modality-specific encoders with language models (Liu et al., 2024a; Zhao et al., 2024). Each

modality's input is first processed by its corresponding encoder, then projected into a shared text embedding space through learnable projection layers, and finally processed by the LLM with text tokens.

The larger size of MLLMs and the need for more data processing power make distributed MLLM training essential. However, heterogeneity in model and data makes balanced MLLM workload distribution more challenging and tackled by recent works (Huang et al., 2024; Zhang et al., 2025; Feng et al., 2025; Jeon et al., 2025). We observe that beyond the first-order disparities in model and data heterogeneity, there are two additional MLLM-specific distributed training challenges that have significant performance implications. First, MLLM training with *frozen* versus *trainable* models results in different computational costs across modules. Model partitioning strategies that do not account for the frozen status of components can lead to suboptimal performance. Second, cross-modality interactions introduce non-causal attention patterns to enable more precise computation of their relationships (Dong et al., 2025; Wang et al., 2025a). While distributing causal attention patterns in LLMs has been extensively studied (Yang et al., 2025; Wang et al., 2025b; Liu et al., 2024b), *efficient distribution of non-causal attention patterns* remains an open challenge.

In this paper, we introduce Cornstarch, an efficient distributed MLLM training framework. Cornstarch transcends the first-order model and data heterogeneity-aware parallelization and uncovers latent higher-order heterogeneities in MLLMs that have not been considered in previous works (Huang et al., 2024; Zhang et al., 2025; Feng et al., 2025; Jeon et al., 2025).

**Frozen status-aware pipeline parallelism (§3.1).** Cornstarch introduces a way to consider the frozen status of MLLM components in model partitioning. We observe that the frozen status of MLLM components can significantly affect the pipeline stage balancing. Existing MLLM approaches do not consider the frozen status in model partitioning (Huang et al., 2024; Zhang et al., 2025; Feng et al., 2025). Even with profiler-based automated approaches that actually measure the backward pass time (Narayanan et al., 2021a; Miao et al., 2022), they cannot account for different computational costs of modules due to the frozen status. We precisely compute the backward pass time based on

---

[1]University of Michigan. Correspondence to: Insu Jang <insujang@umich.edu>.

*Proceedings of the 43rd International Conference on Machine Learning*, Seoul, South Korea. PMLR 306, 2026. Copyright 2026 by the author(s).

[1]https://github.com/cornstarch-org/Cornstarch

the frozen status and the placement of modules to enable accurate pipeline stage balancing.

**Workload balanced context parallelism for MLLMs (§3.2).** Cornstarch introduces novel workload distribution algorithms that balance the computational cost of non-causal attention patterns at both inter-GPU and intra-GPU granularity. At the inter-GPU granularity, we compute the computational cost of each token and distribute them so that GPUs have similar amount of workload as much as possible. However, we notice that even if the GPUs have the same total amount of workload, the workload may not be evenly distributed to compute units within a GPU that leads to imbalance and performance degradation. At the intra-GPU granularity, we shard the workload within each GPU to balance the computational cost across compute units.

We have implemented Cornstarch and conducted extensive evaluations on MLLMs of varying structures, modalities, and sizes. Our evaluation results show that Cornstarch outperforms existing approaches by $2.26\times$ on average ($1.61\times$ – $3.59\times$ across various model sizes) in training throughput.

To summarize, we make the following contributions:

- We identify higher-order heterogeneity-borne challenges in MLLMs that affect the performance of distributed MLLM training.
- We design Cornstarch, a general-purpose distributed MLLM training framework that addresses those challenges.
- Our evaluation shows that Cornstarch surpasses existing approaches by $2.26\times$ on average in training throughput.

## 2. Background and Motivation

We first introduce 4D-parallel distributed LLM training (§2.1). We then enumerate the unique characteristics of MLLMs that challenge existing LLM oriented training paradigms which motivates us to design Cornstarch (§2.2).

### 2.1. 4D Parallelism in LLM Training

Large-scale LLM training is a well-studied topic and leverages four parallelism dimensions – tensor, pipeline, data, and context parallelism – to achieve high training throughput (Narayanan et al., 2021b; Liang et al., 2025; AI, 2024; Rasley et al., 2020; Jiang et al., 2024b). Tensor and pipeline parallelism (TP and PP) partitions the model within each layer and across layers, respectively. Data and context parallelism (DP and CP) partitions data; the former partitions a large batch of sequences into smaller minibatches, while the latter partitions each input sequence into segments. In all parallelization dimensions, balancing the workload across GPUs is important to achieve high throughput. In model partitioning, pipeline stages may have different amount of

computation thus balancing them has been extensively studied (Sun et al., 2024; Narayanan et al., 2021a; Miao et al., 2022; Zheng et al., 2022; Unger et al., 2022). In data partitioning, the amount of workload can be imbalance across data parallel replicas and within each replica due to LLM's causal attention pattern (Ge et al., 2025; Liu et al., 2024b; Wang et al., 2025b; Yang et al., 2025).

### 2.2. Unique Characteristics of MLLMs

MLLMs have unique characteristics that introduce new challenges to the existing 4D parallelism.

**Model and data heterogeneity.** Unimodal LLMs usually contain repeated transformer layers with homogeneous structure, and unimodal text inputs go through the entire model. Unlike unimodal LLMs, MLLMs have multiple modality encoders prior to the LLM with different structures. The way of processing the input data is also different. Modality encoders first process the modality input data that they are responsible for, and then project the output to the LLM. The LLM then embeds the output of all modality encoders and text embedding, and computes the final output.

Model and data heterogeneity of MLLMs introduce imbalance in distributed training, which has been addressed by recent works. For instance, DistMM (Huang et al., 2024), DistTrain (Zhang et al., 2025), Optimus (Feng et al., 2025), and DIP (Xue et al., 2026) disaggregate parallelism by applying different parallelization strategies to modalities to balance heterogeneous workload.

**Using pretrained models with different frozen status.** While unimodal LLMs are usually trained from scratch with randomly initialized parameters, MLLM training typically starts with a pretrained LLM and multiple unimodal encoders to exploit the representative capabilities of the pretrained models (Liu et al., 2024a; Wang et al., 2024; Chen et al., 2024b; AI, 2024). Projectors between the modality encoders and the LLM are newly added and trained to align the embedding spaces of the modality encoders and the LLM. Usually, the modality encoders and the LLM are frozen during MLLM training and only the projectors are trained (Laurençon et al., 2024; Li et al., 2025).

The frozen status of MLLM components significantly alters the amount of computation during the backward pass, as depicted in Figure 1a and Table 1b, invalidating the long-held rule of thumb that *backward passes take roughly twice as long as forward passes* (Narayanan et al., 2021a). When pipeline parallelism is used, lack of considering the frozen status leads to execution time imbalance across pipeline stages. Figure 1a illustrates the impact of frozen status to the balance of pipeline stages. The model is partitioned to 4 pipeline stages to be balanced when all parameters are trainable, however, when the encoder is frozen, the balance

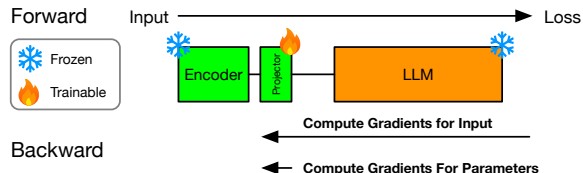

*(a)* The frozen status and the location of layers alters the amount of gradient computation during the backward pass.

*(b)* Forward and backward pass execution time of the VLM with different combination of frozen status. Activation checkpointing is enabled ([Korthikanti et al., 2023](#)).

| | Trainable | | Encoder Time (ms) | | LLM Time (ms) | |
|------|-------|------|-------|--------|--------|--------|
| Enc | Proj | LLM | Fwd | Bwd | Fwd | Bwd |
| $\checkmark$ | $\checkmark$ | $\checkmark$ | 44.43 | 120.57 | 140.39 | 375.93 |
| $\times$ | $\checkmark$ | $\times$ | 44.26 | **0.50** | 138.33 | **284.50** |

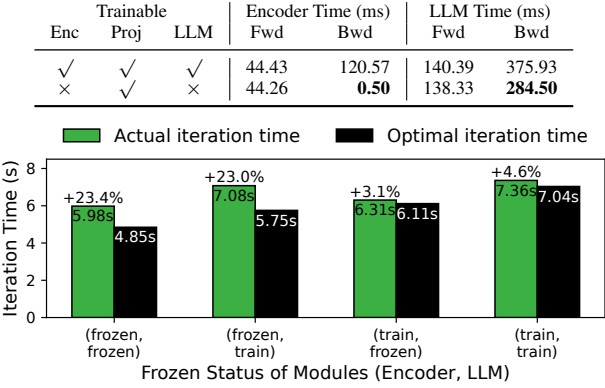

*(c)* Execution time of the VLM with different combination of frozen status using pipeline parallelism on 4 NVIDIA A40 GPUs. The number of microbatch is 64.

*Figure 1.* The impact of frozen status to the backward pass and the balance of pipeline stages. A VLM with Siglip (Encoder) and Llama-3.2 1b (LLM) is used.

between pipeline stages breaks, leading to slower execution.

**Non-causal attention patterns break CP balance.** In LLMs, a token attends to all preceding tokens and itself, forming a lower triangular matrix in the attention matrix, namely causal attention, as illustrated in Figure 2a. Most existing context parallelism works exploit the causal attention pattern ([Wang et al., 2025b](#); [Yang et al., 2025](#)). They partition the sequence into $2 \times$ cp_size, where cp_size is the number of ranks in context parallelism dimension, and the $i$-th rank gets $i$-th and $(2 \times$ cp_size $- 1 - i)$-th chunks. This distribution guarantees balanced workload in causal attention.

However, MLLMs have non-causal attention patterns due to cross-modality interactions, as shown in Figure 2b, where the same distribution is imbalanced. Moreover, the attention pattern is variable depending on the input data, different from LLMs that have fixed causal attention pattern. Statically distributing the workload assuming fixed attention pattern is not applicable to MLLMs.

DCP ([Jiang et al., 2025](#)) is the first work that considers dynamic non-causal context parallelism. However, its de-

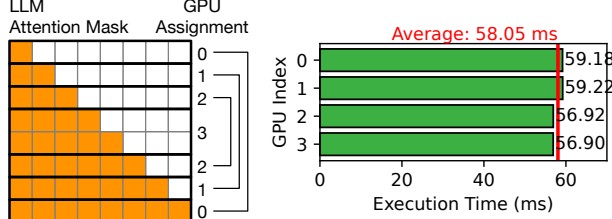

*(a)* Causal attention in LLM can easily be distributed evenly.

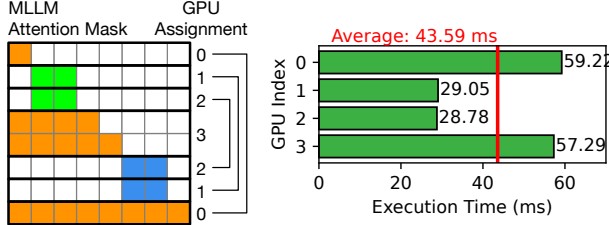

*(b)* Distributing MLLM attention patterns in the same way as in causal attention distribution is imbalanced.

*Figure 2.* Balanced context parallelism optimized for LLMs. It is not applicable to MLLMs.

sign is based on ring context parallelism ([Liu et al., 2024b](#)), which has proven to be inefficient in large scale training and replaced by All-Gather based context parallelism ([Gu et al., 2024](#); [Chu et al., 2025](#)).

## 3. Multimodality-Aware Parallelization

Based on the observation in Section 2, we introduce Cornstarch's MLLM-specific parallelization strategies. We first introduce frozen status-aware pipeline parallelism (§3.1) and then workload-balanced context parallelism (§3.2).

### 3.1. Frozen Status-Aware Pipeline Parallelism

Cornstarch's frozen status-aware pipeline parallelism partitions the model into pipeline stages where the sum of one forward execution time and one backward execution time (1F+1B) is balanced across stages considering the frozen status of the layers.

However, simply considering the frozen status and adopting all-or-nothing approach – add backward pass computation time if trainable, otherwise skip – is not correct either. Even if a layer is frozen, it may still need to backpropagate gradients to the trainable parameters ahead of it. Backward pass computation of a layer $L_l$ consists of two parts: gradient computation for parameters $B_{wl}$ and gradient computation for data $B_{dl}$ ([Qi et al., 2024](#)):

$$B_l = B_{wl} + B_{dl} \tag{1}$$

If trainable parameters are located ahead of a frozen layer, the frozen layer, while it can skip the gradient computation

for its parameters (i.e., $B_w = 0$), needs to *backpropagate input gradients* to the trainable parameters (i.e., $B_d \neq 0$), so that the trainable parameters can update themselves. Then, the backward pass execution time $B_l$ can be computed as:

$$B_l = (B_{wl} \text{ if } f(L_l) \text{ is False else } 0) \tag{2}$$
$$+ (B_{dl} \text{ if } p(L_l) \text{ is True else } 0)$$

where $f(L_l)$ returns the frozen status of the $l$-th layer $L_l$ and $p(L_l)$ returns whether the $l$-th layer $L_l$ has trainable parameters ahead of it. $p(L_l)$ exhibits forward propagation; once it is set to True at some layer that is trainable, all the layers after it need to have $p(L)$ True to backpropagate gradients. Thus, $p(L_l)$ can be computed as:

$$p(L_l) = \begin{cases} \text{True} & \text{if } p(L_{l-1}) \text{ or not } f(L_l) \\ \text{False} & \text{otherwise} \end{cases} \tag{3}$$
$$p(L_0) = \text{not } f(L_0)$$

Checking trainable parameters ahead of the layer should also be done across modalities. If some parameters in the modality encoders are trainable, all layers after it in the same modality as well as the LLM need to backpropagate gradients, since the LLM sits after the modality encoders.

The per-layer cost $T_l = F_l + B_l$ derived from these rules is then used to partition the model into $K$ balanced pipeline stages, minimizing the bottleneck stage cost. Any off-the-shelf partitioning algorithms – such as dynamic programming (Zheng et al., 2022) or divide and conquer (Jang et al., 2023) – can be used to partition the model into pipeline stages, as long as it accepts the per-layer cost $T_l$ as input.

## 3.2. Token Workload-Balanced Context Parallelism

In multimodal context parallelism, many non-causal attention masks can be generated (Li et al., 2023a; Deepmind, 2025; Wang et al., 2025a; Dong et al., 2025), which the existing token distribution for LLMs fails to balance. We observe that to achieve genuine workload-balanced context parallelism, workload distribution across and within GPUs should be considered simultaneously. We call them *inter-GPU* and *intra-GPU* workload balancing and discuss in Section 3.2.1 and Section 3.2.2, respectively.

### 3.2.1. INTER-GPU WORKLOAD BALANCING

Inter-GPU workload imbalance indicates that the amount of workloads distributed to each GPU is not balanced. This is because modern attention implementations introduce variations in the amount of computation per token (Dao et al., 2022; Dao, 2024; Dong et al., 2025). They partition tokens into blocks and skip block computations for efficiency if the corresponding block is completely masked-out. The amount of workloads to compute attention output per query token

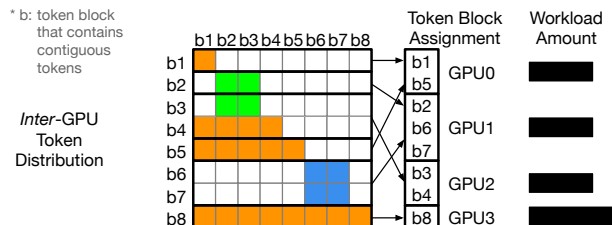

*(a)* Inter-GPU workload balancing. Tokens are distributed across GPUs.

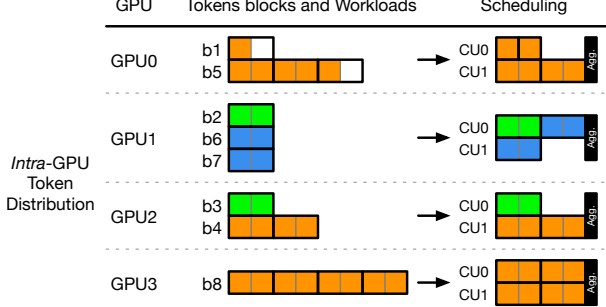

*(b)* Intra-GPU workload balancing. Each token has at most 8 blocks to compute. These blocks are partitioned into subblocks of up to 2 blocks each, creating at most 4 subblocks per token that are scheduled to CUs.

*Figure 3.* Two-step (inter-GPU and intra-GPU) workload balanced context parallelism.

block can be computed by counting the number of colored blocks *rowwise*. In Figure 3a, for example, the workloads of 8 blocks are 1, 2, 2, 4, 5, 2, 2, 8, respectively, which are varied and irregular. We therefore propose a new method of distributing the tokens across GPUs based on the amount of computations, which we call *inter-GPU workload-balanced distribution*.

**Problem formulation.** We first formulate the problem as an integer linear programming (ILP) problem as follows:

$$\begin{aligned} \underset{x, C}{\text{minimize}} \quad & C \\ \text{subject to} \quad & \sum_{g=1}^{G} x_{i,g} = 1, \qquad i = 1, \dots, T, \\ & \sum_{i=1}^{T} W_i \cdot x_{i,g} \leq C, \quad g = 1, \dots, G, \\ & x_{i,g} \in \{0, 1\} \end{aligned} \tag{4}$$

Here, $x_{i,g}$ is a binary decision variable that indicates whether token $i$ is assigned to $g$-th GPU over $G$ GPUs. $W_i$ represents the workload of $i$-th token $x_i$, which can be computed by row-wise sum of unmasked part of the attention mask that needs computation. The linear programming balances workload by minimizing the completion time $C$,

**Algorithm 1** Token workload-balanced context parallelism algorithm.

---

**Input:** Tokens $T$, block size $N_B$, # GPUs $G$, and attention mask $A$
**Output:** Token assignment to GPUs $X_0, X_1, \ldots, X_{G-1}$
$B \leftarrow$ partition $T$ into blocks of size $N_B$
**for** $b \in B$ **do**
  $W_b \leftarrow$ # blocks to compute in attention $A$ for $b$
**end for**
$L \leftarrow \texttt{minheap()}, X \leftarrow \texttt{dict()}$
**for** $g \in 0, \ldots, G-1$ **do**
  `L.heappush(g,0)`
  `X[g] = list()`
**end for**
**for** $b, W_b \in B, W$ **do**
  $g, W[g] \leftarrow$ `L.heappop()`
  `X[g].append(b)`
  `L.heappush(g,W[g] + `$W_b$`)`
**end for**
**return** $X$

---

which is the maximum workload assigned to any GPU.

**Weighted makespan minimization.** For a long sequence, the ILP problem is intractable in real-time during training, thus we adopt the greedy Longest-Processing-Time-First (LPT) algorithm to assign tokens to GPUs in a context parallelism group for fast and efficient distribution (Graham, 1969). Algorithm 1 shows an adapted LPT algorithm that considers the characteristics of parallel accelerators that compute with a large amount of data. We first partition the tokens into blocks of size $N_B$ (e.g., 128). For each token block, we count the number of blocks to compute to measure the workload of the token. If the corresponding attention mask block is full of zeros, the block is skipped. We then use the LPT algorithm to assign the token block to the GPU with the least amount of workload assigned so far.

The longest processing time in the worst case has proven to be $\sum_{i=0}^{T-1} \frac{t_i}{G} + t_{max}$, where $i$-th token's amount of attention computation is $t_i$, total number of tokens $T$, and the number of GPUs $G$ (Graham, 1969). As $T$ increases, $\sum \frac{t_i}{G}$ dominates the processing time, and it is getting closer to the perfectly balanced distribution. It requires $O(GTlogT)$ time complexity, where $TlogT$ is consumed by sorting the tokens in descending order of their workloads.

### 3.2.2. INTRA-GPU WORKLOAD BALANCING

Even with inter-GPU workload balanced distribution, which evenly distributes the *total amount of computation* across GPUs, architectural characteristics of GPUs and implementation of attention can still lead to imbalanced execution when the jobs are dispatched to compute units (CUs).

Revisiting modern attention implementations (Milakov & Gimelshein, 2018; Rabe & Staats, 2022; Dao et al., 2022; Dao, 2024; Dong et al., 2025), they are designed to avoid unnecessary memory accesses as much as possible. CUs use online softmax algorithm and compute the final attention output of a single query token block by keeping the intermediate output in the cache and iterating over the entire key and value blocks in a single kernel. This minimizes the number of memory accesses by not writing intermediate variables to global memory.

However, assigning attention computation of a block *as a whole* to a CU introduces imbalanced amount of workload across CUs. In Figure 3b, for example, b1 and b5 assigned to GPU0 are executed in parallel on CU0 and CU1, respectively. While the amount of computation of b1 (1 block) and b5 (5 blocks) are extremely different, computing b5 cannot be parallelized across CUs; thus, CU0 has to wait for CU1 to finish before proceeding to the next kernel execution.

We observe that the idea of blockwise parallel attention, which was originally designed to parallelize attention across multiple accelerators, can also be used to balance the workload across CUs in a single GPU (Liu & Abbeel, 2023). We adopt it for intra-GPU workload balancing, where the attention computation of a single set of query tokens is split into multiple subblocks and scheduled in parallel. Figure 3b, for example, partitions attention computations to subblocks of size 2. Each kernel computes partial attention output for a single subblock and writes it to local memory. Then, we launch an additional aggregation kernel that gathers the local attention outputs and computes the final attention output for all query blocks. Unlike the original attention computation, which writes the final attention output to memory, our output is local attention output per subblock that needs to be aggregated. The size of blocks (e.g., 2 subblocks per block in Figure 3b) affects the performance; with smaller subblocks, workloads are more balanced, but more local outputs should be written to the global memory, and then read again for aggregation. Large subblocks, on the other hand, have less overhead of aggregation but more imbalance. We empirically find that using $16-32$ subblocks per block achieves good performance.

## 4. Implementation

Cornstarch is implemented in around 26k new Python SLOC on top of PyTorch 2.6.0 (Paszke et al., 2019), HuggingFace Transformers 4.51.0 (Wolf et al., 2020), and Colossal-AI 0.4.6 (Li et al., 2023b). Cornstarch's model partitioning, scheduling, execution, communication, and checkpointing are implemented upon Colossal-AI interface. Cornstarch supports various model families and model sizes so that users can train mode than 10,000 different combinations of MLLMs. See Appendix B for the list of supported models.

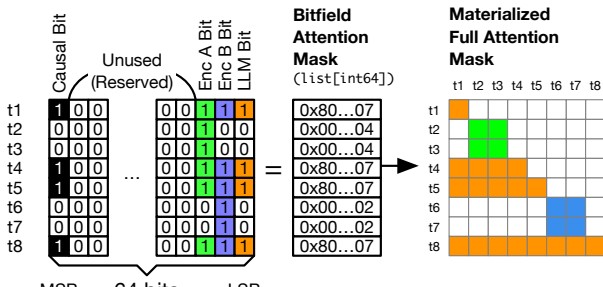

*Figure 4.* Bitfield attention mask representation.

## 4.1. Implementation of Pipeline Stage Partitioning

Cornstarch assigns each layer a per-layer cost from Section 3.1 and partitions the model into $K$ contiguous pipeline stages. The partitioner minimizes the bottleneck stage cost – the 1F+1B time of the slowest stage – which is the standard objective for pipeline stage balancing (Zheng et al., 2022; Jang et al., 2023). Cornstarch solves this contiguous partitioning problem with dynamic programming (Zheng et al., 2022); the search space is small in practice (hundreds of layers and up to tens of stages), so partitioning completes in negligible time relative to training. Let $w(j, i) = \sum_{\ell=j}^{i-1} T_\ell$ be the cost of the contiguous segment from layer $j$ to $i-1$. $X[i][k]$ is the optimal value when the first $i$ layers are partitioned into $k+1$ stages:

$$X[i][0] = w(0, i) \tag{5}$$
$$X[i][k] = \min_{j \in [k, i)} \max\big(X[j][k-1], \, w(j, i)\big)(k > 0) \tag{6}$$

The recurrence tries every split point $j$: the bottleneck is the larger of the optimal bottleneck for the prefix of length $j$ with $k$ stages and the cost $w(j, i)$ of the last stage. Cornstarch records the argmin $j$ at each subproblem; the optimal bottleneck cost is $X[N][K-1]$. To obtain the partition, Cornstarch backtracks from $(N, K-1)$: the stored $j$ marks where the last stage begins, so layers $j$ through $N-1$ form the final stage, and the algorithm recurses on the prefix subproblem $(j, K-2)$. Repeating until $k = 0$ yields all $K$ contiguous stage boundaries.

## 4.2. Implementation of MLLM Attention

Cornstarch implements *bitfield attention* in Triton (Tillet et al., 2019; OpenAI, 2021) for high performance non-causal attention execution. Naive attention implementation computes attention scores for all tokens, and then applies the attention mask as a whole to the attention scores. This is inefficient as it requires high memory bandwidth and is not parallelizable. Bitfield attention mask is a sparse representation of the attention mask to represent multimodal interactions into attention patterns efficiently. Full attention mask is a very large 4D tensor (batch × # heads × sequence

length × sequence length), which needs too much memory for long sequences. Bitfield attention mask is a 2D 64-bit integer tensor (batch × sequence length), where each bit represents which modalities the token at that position needs to attend to. Figure 4 shows an example of a bitfield attention mask. We assign bits from the least significant bit (LSB) to the most significant bit (MSB) to the modality encoders and the LLM. The LSB (1st index) is assigned to the LLM, and 2nd and 3rd bits are assigned to the modality encoders A and B, respectively, for example. The most significant bit (64th index) is reserved for causal bit; when this bit is set to 1, the token attends to all of its previous tokens. For example, tokens `t2`, `t3` are tokens from the encoder A, thus have 2nd LSB set to 1. As it does not have causal bit, it attends to the other tokens only from the encoder A. `t4`, `t5`, however, are text tokens with causal bit and all modality bits set to 1. Thus it can attend to all modality tokens, but only previous tokens. Cornstarch bitfield attention implementation is compatible with context parallelism (§4.3), while standard FlashAttention does not natively support.

## 4.3. Implementation of Context Parallelism

There are various ways of implementing context parallelism: all-to-all (Jacobs et al., 2023), ring attention (Liu et al., 2024b; Jiang et al., 2025), and All-Gather based (AI, 2024; Chu et al., 2025), etc. Cornstarch implements the SOTA All-Gather based context parallelism implementation. This implementation gathers all keys and values of all tokens and compute row-wise attention for local queries. Overlapping communication and computation is done in the head dimension; while GPUs compute attention for one or a few heads, it transfers keys and values for the next head(s). This simplifies Algorithm 1 in computing per-token workload. If we adopt P2P ring attention, it would have been more complicated to compute per-token workload as it requires to recompute the amount of workloads every round.

## 5. Evaluation

In this section, we evaluate Cornstarch and show its effectiveness in training MLLMs. Our key results are:

- Cornstarch achieves $2.26\times$ higher end-to-end training throughput on average for MLLM training (§5.2).
- Frozen status-aware pipeline parallelism partitions MLLMs more effectively by considering the frozen status and provides up to $2.46\times$ faster iteration time in MLLMs (§5.3).
- Workload-balanced context parallelism distributes tokens more evenly across GPUs and within a single GPU, which improves the performance of attention execution by up to $1.18\times$ (§5.4).

*Table 1.* Modality (LLM, vision, and audio) configurations.

| Model Arch. | Model Size | # Layers | Hidden Size | # Params |
|---|---|---|---|---|
| Llama-3 (LLM) | Small | 16 | 2048 | 1b |
| | Medium | 32 | 4096 | 8b |
| | Large | 64 | 5120 | 32b |
| Qwen2 Vision | Small | 32 | 1280 | 0.6b |
| | Medium | 48 | 2560 | 3.9b |
| | Large | 64 | 3840 | 11.6b |
| Phi4 Audio | Small | 24 | 1024 | 0.5b |
| | Medium | 32 | 3072 | 3.4b |
| | Large | 48 | 5120 | 12.4b |

## 5.1. Experimental Setup

**Testbed.** We run our evaluation workloads in a GPU cluster with 6 nodes, each with four NVIDIA A40-48GB GPUs and a NVIDIA Mellanox ConnectX-6 200Gbps Infiniband adaptor (total 24 GPUs). The four GPUs in a node are connected in pairs using NVLink and connected to the node via PCIe 4.0.

**Baselines.** We set the baselines as follows:

1. *FSDP*: FSDP is widely used in distributed MLLM training thanks to its ease of use (Chen et al., 2024b; Hong et al., 2024; Liu et al., 2023; 2024a). It shards parameters and distributes them across all GPUs to reduce memory footprint. Parameters are temporarily gathered for computation and then sharded again. We use FSDP2, which offers higher performance (Liang et al., 2025).

2. *Megatron\**: Megatron-LM extends LLM pipeline parallelism to MLLMs by adding a vision encoder as the first pipeline stage (NVIDIA, 2024). We chose Megatron-LM as a representative of the existing LLM-optimized 4D parallelization.

**Training data.** We use a synthetic dataset for evaluation. Each sample consists of 1k text tokens, a 1280x720 image, and a 2-minute audio clip. Image tokens and audio tokens are projected into the text embedding space after being processed by the corresponding modality encoder. We use a global batch size of 48. FSDP uses a mini-batch size of 2, while microbatch size 4 is used in Megatron\* and Cornstarch with pipeline parallelism. The attention patterns are described in Appendix C.

**Model configurations.** We evaluate various MLLM configurations created by combining two modality encoders (vision and audio) and an LLM, each selected from the sizes listed in Table 1. The modality encoders are merged into a single module which processes both modalities. We freeze the merged modality encoder and the LLM and only train the projector modules. An MLLM configuration is denoted by suffixes representing the sizes (S, M, L) of its

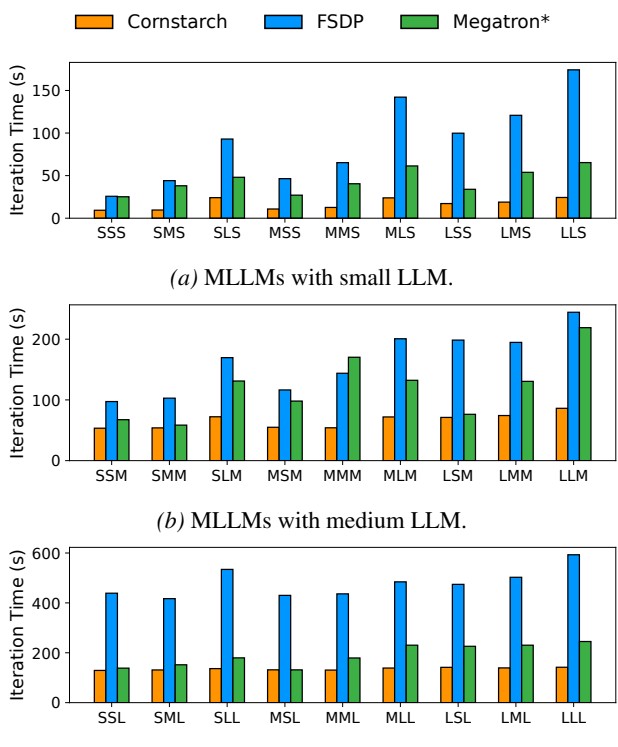

*(a)* MLLMs with small LLM.

*(b)* MLLMs with medium LLM.

*(c)* MLLMs with large LLM.

*Figure 5.* End-to-end performance comparison of Cornstarch and baselines with various model configurations.

vision encoder, audio encoder, and LLM, respectively (e.g., MLLM-SML combines a small vision encoder, a medium audio encoder, and a large LLM).

## 5.2. End-to-End Performance

We first evaluate Cornstarch against the baselines in terms of end-to-end training iteration time and show the results in Figure 5. When models are small enough (e.g., MLLM-SSS in Figure 5a), FSDP shows reasonably good performance. However, as model size increases, FSDP's performance drops significantly due to intensive communication overhead. For Megatron\*, its current limitations cause a large imbalance in pipeline stages due to the lack of frozen status awareness. This is observed especially well when the modality encoders are relatively larger than LLMs (e.g., MLLM-LLS in Figure 5a or MLLM-LLM in Figure 5b).

Cornstarch shows the best performance across all model configurations. It chooses better modality parallelism, allocates pipeline stages to the encoders and the LLM based on their frozen status and their placement, and balances the attention computation on the fly. We discuss the performance of Cornstarch in detail in the subsequent sections. Overall, Cornstarch outperforms the baselines 2.26× on average (3.36× vs FSDP and 1.62× vs Megatron\*).

*Table 2.* Model forward and backward execution time breakdown parallelized with and without frozen status awareness.

| Model | Frozen Aware | Per-Stage Fwd (ms) | | Per-Stage Bwd (ms) | | Iter. Time (s) | Impr. (×) |
|---|---|---|---|---|---|---|---|
| | | Enc | LLM | Enc | LLM | | |
| SSS | √ | 301.61 | 149.40 | 1.04 | 518.43 | 21.81 | 1.18x |
| | × | 207.13 | 296.25 | 0.86 | 1032.63 | 25.67 | - |
| MMS | √ | 903.86 | 102.61 | 2.72 | 346.54 | 40.34 | 1.02x |
| | × | 635.56 | 297.62 | 1.19 | 1032.12 | 41.21 | - |
| LLS | √ | 1240.50 | 298.37 | 1.41 | 1029.98 | 66.14 | 1.00x |
| | × | 1259.13 | 297.92 | 1.15 | 1030.14 | 66.20 | - |
| SSM | √ | 464.98 | 331.43 | 3.53 | 3017.01 | 66.56 | 1.11x |
| | × | 388.33 | 388.73 | 2.30 | 4030.74 | 73.94 | - |
| MMM | √ | 2330.90 | 273.89 | 2.09 | 2418.27 | 70.37 | **2.46x** |
| | × | 712.72 | 1159.76 | 1.60 | 12113.67 | 173.01 | - |
| LLM | √ | 2199.91 | 376.11 | 4.15 | 4023.22 | 87.43 | 2.03x |
| | × | 1403.97 | 1161.76 | 1.56 | 12109.73 | 177.39 | - |
| SSL | √ | 773.10 | 741.17 | 3.55 | 6309.11 | 138.45 | 1.00x |
| | × | 774.19 | 708.62 | 3.84 | 6306.18 | 137.62 | - |
| MML | √ | 2280.58 | 705.73 | 3.11 | 6311.23 | 138.97 | 1.30x |
| | × | 1015.60 | 1154.85 | 2.78 | 10546.34 | 180.28 | - |
| LLL | √ | 5316.08 | 736.05 | 4.00 | 6315.46 | 143.76 | 1.72x |
| | × | 1597.36 | 1686.28 | 3.06 | 15878.58 | 247.79 | - |

## 5.3. Impact of Frozen Status-Aware Pipeline Parallelism

We parallelize the models with frozen status-aware pipeline parallelism and compare the performance with the same models but parallelized without frozen status awareness. Table 2 presents the results. For brevity, we only show a few model configurations with encoders being colocated.

Without frozen status-awareness, partitioning is done based on the assumption of all parameters being trainable, which tries to minimize variance of forward time across pipeline stages. For example, MLLM-LLL, the frozen status-unaware partitioning partitions the modality encoders and the LLM to have similar forward execution time (∼ 1600ms). However, gradient computations for the frozen encoders and the LLM are skipped, their backward execution time is significantly different (3.06ms and 15878.58ms), breaking the balance between pipeline stages.

With frozen status-awareness, the partitioning is balanced based on the forward execution time plus the backward execution time (5320.08ms and 7051.51ms, respectively), decreasing pipeline bubbles. We also observe a few exceptions: MLLM-LLS and MLLM-SSL. These are the most extreme cases in model size distinction, thus even assigning maximum number of pipeline stages to the large module is not enough to balance the pipeline stages. In other cases, frozen status-aware pipeline parallelism assigns workloads more evenly across pipeline stages, which improves the overall performance by up to 2.46×.

## 5.4. Impact of Workload-Balanced Context Parallelism

This section evaluates how Cornstarch's workload-balanced context parallelism (§3.2) distributes non-causal attention

*Table 3.* Execution time of a single attention layer and entire LLM with 64k sequence length using various context parallelization policies.

| Time (ms) (Impr. (×)) | | Causal CP | Inter-GPU Balance Only | Intra-GPU Balance Only | Cornstarch |
|---|---|---|---|---|---|
| LLM-S | Attn | 243.44 (-) | 255.59 (0.95x) | 225.73 (1.08x) | 204.95 **(1.19x)** |
| | Model | 5541.25 (-) | 5665.77 (0.98x) | 5250.40 (1.06x) | 4856.60 (1.14x) |
| LLM-M | Attn | 460.13 (-) | 487.44 (0.94x) | 440.86 (1.04x) | 417.31 (1.10x) |
| | Model | 24534.50 (-) | 25389.74 (0.97x) | 23712.89 (1.03x) | 22815.79 (1.08x) |
| LLM-L | Attn | 568.18 (-) | 610.67 (0.93x) | 558.56 (1.02x) | 551.60 (1.03x) |
| | Model | 77378.44 (-) | 79671.34 (0.97x) | 75055.69 (1.03x) | 74864.71 (1.03x) |

execution well. We run LLMs with 64k sequence length, where the attention mask is simulated to represent a mixed of multiple modalities. See Appendix D for results with different sequence lengths.

Table 3 shows the results of a single attention layer and the entire LLM with various context parallelization policies. We only show the results of LLM-L, as the same patterns are observed in other model sizes. Cornstarch shows the best performance, outperforming the existing causal context parallelism optimized for LLMs by up to 1.18×. Intra-GPU workload balancing also shows improvement. Even with additional overheads from aggregation, parallelizing attention subblocks within a single GPU effectively removes tail latency caused by stragglers.

Surprisingly, however, balancing workload distribution only at a token level (inter-GPU balancing only) does not provide performance improvement. To understand this, we further perform CU activity analysis of a single attention layer, depicted in Figure 6. Severe downward spikes are observed in both causal context parallelism (Figure 6a) in inter-GPU only balance context parallelism (Figure 6b). The spikes happen at the end of every attention head computation. This is because attention kernel for the next head cannot be launched until the in-flight attention kernel for the current head is entirely finished, leaving CUs inactive. Intra-GPU balancing fundamentally solves this problem by distributing workloads of attention computation of each single block in finer granularity across CUs, showing higher CU activity (Figure 6c). Still, only balancing Intra-GPU workloads does not balance the total amount of workloads across GPUs; some GPUs become idle much earlier while others are busy, reducing overall utilization. Combining

---

[1] A simpler alternative solution is to use multiple CUDA streams to overlap attention computations across heads. However, our experiments shows that it does not solve the CU underutilization problem. See Appendix E.

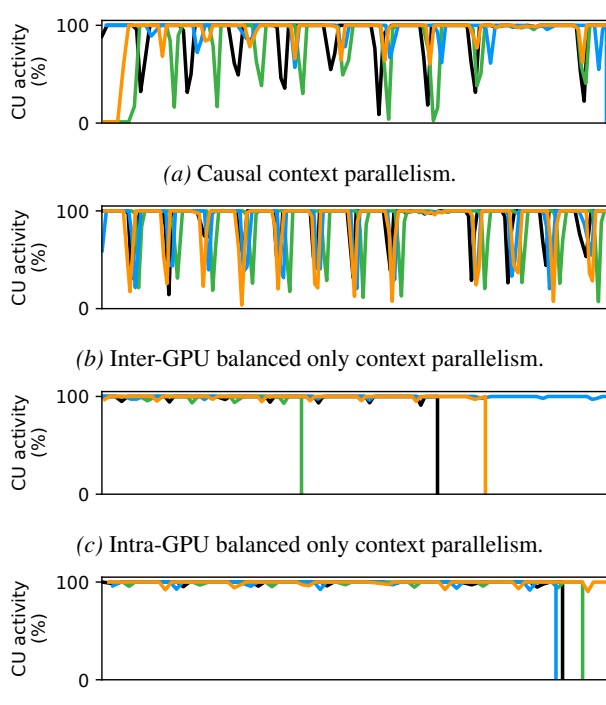

*(a)* Causal context parallelism.

*(b)* Inter-GPU balanced only context parallelism.

*(c)* Intra-GPU balanced only context parallelism.

*(d)* Cornstarch context parallelism.

*Figure 6.* CU activity analysis with various context parallelization policies running a single attention layer of LLM-L. Each line represents one GPU.

*Table 4.* Cornstarch and DCP comparison using Lambda attention pattern. Time is in milliseconds.

| Seq length | 32k | 64k | 128k | 256k |
|---|---|---|---|---|
| Cornstarch | 177.41 | 143.56 | 375.49 | 1193.32 |
| DCP | 46.36 | 124.72 | 417.07 | 1545.83 |

inter- and intra-GPU balancing, Cornstarch achieves the best performance (Figure 6d).

### 5.5. Comparison with DCP

While DCP (Jiang et al., 2025) is not designed for MLLMs, its capabilities can be adapted to MLLMs, representing MLLM-specific attention patterns. We compare Cornstarch with DCP in terms of single attention layer performance and end-to-end training time on 16 GPUs. We use 4 attention patterns demonstrated in the DCP paper for comparison.

Table 4 shows the result of a single attention layer with lambda attention pattern. See Appendix F for other patterns' result. With shorter sequence length, DCP shows better performance than Cornstarch. This is because Cornstarch requires enough sequence length to fully hide the communication overhead. With longer sequence length, however, DCP rather struggles to hide the communication overhead, since it relies on ring context parallelism, which

*Table 5.* End-to-end training time on Cornstarch and DCP.

| Seq Length | Time (s) | Causal | Lambda | Causal Blockwise | Shared Questions |
|---|---|---|---|---|---|
| 32k | Cornstarch | 5.7 | 5.8 | 10.5 | 5.7 |
| | DCP | 26.5 | 49.0 | 42.0 | 46.5 |
| 128k | Cornstarch | 37.2 | 37.4 | 29.0 | 33.9 |
| | DCP | 46.5 | 107.1 | 100.8 | 102.4 |

is inefficient in training with long sequence lengths or large number of context parallel degrees, while Cornstarch all-gather based context parallelism gets more benefits from longer sequences.

Table 5 shows a full iteration time of Llama3-1b model (planning + 1 forward + 1 backward). DCP suffers from significant planning overhead to find the optimal computation schedule. Cornstarch's heuristic token assignment is efficient, finishing planning in less than 1 second in all cases, outperforming DCP.

## 6. Related Work

**4D parallelism.** Large-scale LLM training combines tensor, pipeline, data, and context parallelism to scale model and sequence length (Narayanan et al., 2021b; AI, 2024; Jiang et al., 2024b). Each dimension has been optimized for pipeline stage balancing and causal context parallelism (Narayanan et al., 2021a; Wang et al., 2025b; Gu et al., 2024). However, these methods assume homogeneous transformers and fixed causal attention, mostly focusing on LLMs, and do not address MLLM-specific heterogeneity, frozen status, or non-causal attention.

**Distributed multimodal training.** MLLM training commonly uses FSDP (Zhao et al., 2023; Liu et al., 2023), but it scales poorly for large models in large clusters (Liang et al., 2025). DistMM (Huang et al., 2024), DistTrain (Zhang et al., 2025), Optimus (Feng et al., 2025), and DIP (Xue et al., 2026) address heterogeneous modality modules, yet overlook frozen-status effects on pipeline balance and non-causal context parallelism.

## 7. Conclusion

In this paper, we presented Cornstarch, a multimodality-aware distributed MLLM training framework. Cornstarch addresses higher-order challenges arising from model and data heterogeneity in MLLM training. We introduce frozen status-aware pipeline parallelism that balances the computational cost of MLLM pipeline stages. We also introduce workload balanced context parallelism which computes the amount of workloads both in intra-GPU and inter-GPU. Cornstarch provides $2.26\times$ speedup over the state-of-the-art distributed MLLM training frameworks on average.

## Acknowledgements

We thank the ICML reviewers and members of SymbioticLab for their helpful discussions and feedback. This work was supported in part by NSF grants CCF-2450085, CCF-2327011, CCF-2504995, CNS-2106184, CNS-2535540, and CNS-2406598 and by grants from Cisco, Ford, Mozilla Foundation, and Laude Institute.

## Impact Statement

This work improves the efficiency of distributed MLLM training. Considering the growing demand for multimodal LLMs, our work is expected to have a significant time and energy savings in training large multimodal models.

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

*Listing 1.* Cornstarch APIs for distributed MLLM training.

```
1   # Load unimodal models
2   vis = SiglipVisionModel.from_pretrained("...")
3   aud = WhisperEncoder.from_pretrained("...")
4   llm = LlamaForCausalLM.from_pretrained("...")
5
6   # Create an MLLM with modality information
7   mllm = MultimodalModule(
8     encoders = {
9       "vision": EncoderModule(vis, proj="mlp"),
10      "audio": EncoderModule(aud, proj="linear"),
11      # ... more encoders
12    },
13    language_model = llm,
14  )
15
16  # Define parallel spec per modality
17  # either by manually or by automatically
18  vis_spec = ParallelSpec(...)
19  aud_spec = ParallelSpec(...)
20  llm_spec = ParallelSpec(...)
21
22  # Parallelize the MLLM
23  torch.distributed.init_process_group(...)
24  dist_mllm = MultimodalParallelModule(
25    mllm,
26    modality_parallelism="parallel",
27    encoder_specs={
28      "vision": vis_spec,
29      "audio": aud_spec,
30      # ... more encoders
31    },
32    language_model_spec=llm_spec,
33    num_microbatches=...,
34    microbatch_size=...,
35  )
36
37  # Run distributed training of MLLM
38  for batch in dataloader:
39    output = dist_mllm.execute(batch)
40    optimizer.step()
41    optimizer.zero_grad()
```

*Table 6.* Supported models in Cornstarch.

| Modality | Model Names |
|---|---|
| LLM | Llama (3, 4) (AI, 2024), Mistral (Jiang et al., 2023), Mixtral (Jiang et al., 2024a), Gemma (1, 2) (Deepmind, 2024a;b; 2025), Qwen (2, 2.5, 3) (Yang et al., 2024), Phi (3, 4) (Microsoft, 2024), InternLM2 (Cai et al., 2024) |
| Vision Encoder | CLIP (Radford et al., 2021), Dinov2 (Oquab et al., 2024), Siglip (Zhai et al., 2023), EvaCLIP (Sun et al., 2023), Pixtral (Agrawal et al., 2024), Qwen2Vision (Wang et al., 2024) |
| Audio Encoder | Whisper (Radford et al., 2023), Qwen2Audio (Chu et al., 2024), Phi4Audio (Abdin et al., 2024) |

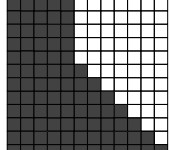 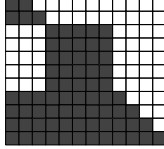 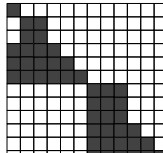

*(a)* Encoder outputs prepended. Early VLMs used (Liu et al., 2023).

*(b)* Encoder outputs embedded. Modern VLMs use a special token such as <image> to specify the location of modality tokens.

*(c)* Multimodal packing. Several sequences with different attention patterns can be packed for more efficient processing.

*Figure 7.* Various attention masks used in MLLM training.

## C. Attention Patterns

We use three different types of attention patterns in our evaluation. Figure 7 illustrates these patterns. Modality tokens can either be prepended or embedded, and several multimodal sequences can be packed into a single long sequence. While the number of tokens for each modality is fixed as specified in Section 5.1, the placement of the modality data is randomly assigned.

## D. Workload-Balanced Context Parallelism

Figure 8 shows context parallelism results with smaller sequence lengths than 64k. Similar patterns as in Table 3 are observed. Intra-GPU balance balances workloads of long sequences across SMs within each GPU. While inter-GPU balance, if applied alone, is worse than context parallelism optimized for causal attention, it provides further optimized performance when combined with intra-GPU balance.

## E. Context Parallelism Using Multiple Streams

Using multiple streams in CUDA can improve performance by allowing concurrent execution of multiple attention head computations. It does improve performance by overlapping attention head computations in the middle of each attention layer as presented in Table 7. However, in Figure 9, spikes are still observed at the end of every attention iteration, as

## A. Cornstarch Programming Interface

Listing 1 shows the programming interface of Cornstarch. MultimodalModule is a wrapper class that contains the modality encoders and the LLM that can be executed standalone without parallelization specifications (line 7). Cornstarch accepts parallelization specifications for each modality encoder and the LLM (line 18 to line 20). The parallelization specifications are passed to MultimodalParallelModule, which is a wrapper class that contains the specification and more hyperparameters required for distributed training (line 24). After creating a distributed MLLM, users can call execute method to run the training (line 39).

## B. Supported Models

Table 6 lists the supported models in Cornstarch at the time of writing. However, Cornstarch is not limited to these models; those in the table are just the ones we have tested.

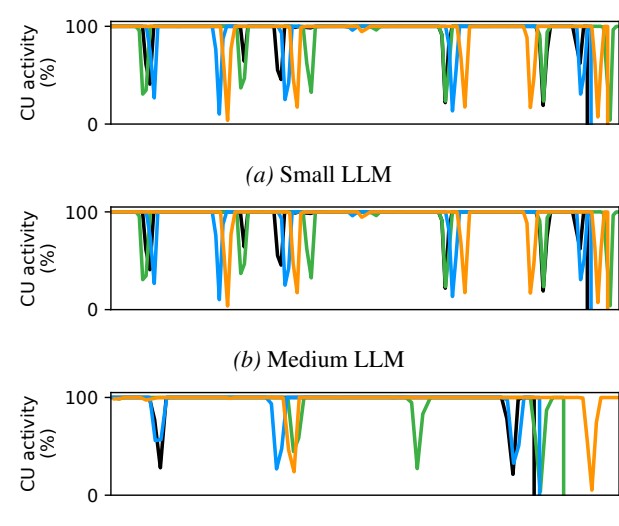

*(a)* Small LLM

*(b)* Medium LLM

*(c)* Large LLM

*Figure 9.* CU activity analysis with inter-GPU balancing + multiple CUDA streams.

*(a)* Workload-balanced context parallelism with 32k sequence

| Time (ms) | | Causal CP | Inter-GPU Balance Only | Intra-GPU Balance Only | Cornstarch |
|---|---|---|---|---|---|
| LLM-S | Attn | 66.00 | 74.44 | 57.65 | 57.32 |
| | Model | 1858.73 | 1976.68 | 1731.12 | 1705.38 |
| LLM-M | Attn | 113.37 | 127.46 | 111.17 | 105.16 |
| | Model | 8345.84 | 8789.94 | 8213.21 | 8029.79 |
| LLM-L | Attn | 148.75 | 159.39 | 141.85 | 143.28 |
| | Model | 29372.62 | 30234.54 | 28767.26 | 28518.80 |

*(b)* Workload-balanced context parallelism with 16k sequence

| Time (ms) | | Causal CP | Inter-GPU Balance Only | Intra-GPU Balance Only | Cornstarch |
|---|---|---|---|---|---|
| LLM-S | Attn | 23.54 | 24.26 | 17.61 | 18.23 |
| | Model | 800.29 | 802.07 | 691.22 | 705.14 |
| LLM-M | Attn | 40.01 | 40.72 | 35.35 | 34.47 |
| | Model | 3682.71 | 3700.22 | 3467.52 | 3491.53 |
| LLM-L | Attn | 48.02 | 50.77 | 44.74 | 41.41 |
| | Model | 13089.76 | 13184.58 | 12677.96 | 12628.59 |

*Figure 8.* Workload-balanced context parallelism with different sequence lengths.

*Table 7.* Model execution time with inter-GPU balancing + using multiple CUDA streams.

| Time (ms) | Causal CP | Inter-GPU Balance + Multistreams | Cornstarch |
|---|---|---|---|
| LLM-S | 5541.25 | 5294.12 | 4856.60 |
| LLM-M | 24534.50 | 23794.29 | 22815.79 |
| LLM-L | 77378.44 | 76457.72 | 74864.71 |

*Table 10.* Shared questions.

| Seq length | 32k | 64k | 128k | 256k |
|---|---|---|---|---|
| Cornstarch | 199.99 | 179.59 | 351.41 | 1190.41 |
| DCP | 39.79 | 120.74 | 374.99 | 1358.43 |

We compare Cornstarch with DCP in terms of single attention layer performance with three different attention patterns demonstrated in the DCP paper (Jiang et al., 2025): causal, causal blockwise, and shared question, as in Table 8, Table 9, and Table 10. All time is in milliseconds.

the last attention head computation cannot be overlapped with the next head.

# F. Comparison with DCP

*Table 8.* Causal.

| Seq length | 32k | 64k | 128k | 256k |
|---|---|---|---|---|
| Cornstarch | 211.51 | 158.15 | 380.88 | 1189.57 |
| DCP | 54.63 | 149.18 | 503.56 | 1861.39 |

*Table 9.* Causal blockwise.

| Seq length | 32k | 64k | 128k | 256k |
|---|---|---|---|---|
| Cornstarch | 228.36 | 124.72 | 333.91 | 1002.70 |
| DCP | 41.13 | 104.87 | 316.93 | 1083.57 |

