# OpenReview forum: "Efficient Distributed MLLM Training with Cornstarch"
_ICML.cc/2026/Conference — ICML 2026 regular_

### Official Review · Reviewer_EnZh · 2026-03-05

**Soundness:** 3
**Presentation:** 3
**Significance:** 3
**Originality:** 3
**Overall Recommendation:** 4
**Confidence:** 3

**Summary:**

This paper proposes ModalGlue, a distributed training framework tailored for multimodal large language models (MLLMs), which integrate heterogeneous modality encoders with large language models to process diverse modalities such as vision and audio. The goal is to improve distributed training efficiency for MLLMs whose heterogeneous structures and non-causal attention patterns create workload imbalance across GPUs. The framework introduces two key techniques: (1) frozen status-aware pipeline parallelism, which partitions pipeline stages by modeling the forward and backward computational costs while accounting for frozen and trainable modules, and (2) workload-balanced context parallelism, which distributes attention computation across GPUs and compute units to address irregular workloads caused by non-causal multimodal attention patterns. Experimental results across various MLLM configurations show that ModalGlue improves training throughput compared with existing distributed training frameworks such as FSDP and Megatron*, achieving up to 2.26× speedup.

**Compliance With Llm Reviewing Policy:**

Affirmed.

**Final Justification:**

The authors have addressed my concerns.

**Key Questions For Authors:**

- Could the authors provide additional discussion or empirical comparisons with OrchMLLM [1] or other recent MLLM training frameworks under similar hardware and data settings?
- Could the authors further explain the questions in weaknesses, particularly with respect to the novelty?

[1] Zheng Y, Xiao B, Shi L, et al. Orchestrate multimodal data with batch post-balancing to accelerate multimodal large language model training[J]. arXiv preprint arXiv:2503.23830, 2025.

**Limitations:**

yes

**Strengths And Weaknesses:**

### Strengths ###
- 1. The paper clearly identifies overlooked “higher-order” scalability bottlenecks specific to MLLMs, including the performance impact of freezing pretrained encoders/decoders and the irregular computation patterns of non-causal, cross-modal attention.
- 2. ModalGlue’s frozen status-aware pipeline partitioning is conceptually well-motivated and goes beyond standard profile-based approaches, leading to demonstrable efficiency gains.
- 3. Claims are well-substantiated with strong empirical data. The overall writing is clear and structured, with key technical details included both in body and appendices.

### Weaknesses ###
- 1. While the paper frames its contributions around “higher-order heterogeneity,” most techniques appear to be engineering refinements of existing parallelization strategies rather than fundamentally new distributed training paradigms (novelty limitation).
- 2. Absence of measurement or analysis of system resource usage beyond wallclock iteration time: e.g., memory usage and energy efficiency.
- 3. The experiments are conducted using synthetic multimodal data rather than real-world multimodal training datasets. As a result, it is unclear whether the observed workload characteristics and performance improvements would generalize to realistic MLLM training workloads.

---

> ### Author Rebuttal · Authors · 2026-03-30
>
> We sincerely thank the reviewer for constructive feedback. We provide clarifications and additional results addressing each concern. Due to limited space, all results are small vision encoder + medium LLM on 4 nodes with 4*A40 each. Projector is always trainable. All time is in ms.
>
> ### Contributions are Engineering Refinement
>
> Our contributions are not engineering refinement. This is like saying finding a new, better algorithm for the same problem is an engineering refinement. We discover characteristics that have long been neglected and drastically improve overall throughput, and we do propose a new training paradigm that frozen status and attention patterns should be considered for optimal distributed training.
>
>
> ## System Resource Analysis (vs DCP)
>
> We compare ModalGlue against DCP in terms of memory consumption on a single attention layer with 64 heads, 8 kv heads, and head_dim=128. For attention patterns, we implement 4 different attention patterns introduced in DCP. All memory consumption is in MB. Due to limited space, we only show 256k result in this comment. But ModalGlue outperforms DCP in all cases.
>
> - 256k
>
> | Memory | Causal | Lambda | Blockwise | Shared Question |
> |:-----------------------:|:------:|:------:|:---------:|:---------------:|
> |        DCP (base)       |  320.0 |  320.0 |     320.0 |           320.0 |
> |        DCP (peak)       | 3749.9 | 3808.9 |    3572.2 |          3643.2 |
> |     ModalGlue (base)    |  341.0 |  341.0 |     341.0 |           341.0 |
> |     ModalGlue (peak)    | 3155.0 | 3155.0 |    3155.0 |          3155.0 |
>
> Base memory consumption difference is because of the bitfield attention mask, because DCP frees the attention mask after compiling to generate blocks and schedule. Note that the attention mask is shared across multiple layers, thus memory overhead will be amortized. More importantly, in all cases, ModalGlue reports lower peak memory consumption than DCP.
>
> ## System Resource Analysis (vs DCP)
>
> We compare ModalGlue against DCP in terms of memory consumption on a single attention layer with 64 heads, 8 kv heads, and head_dim=128. For attention patterns, we implement 4 different attention patterns introduced in DCP. All memory consumption is in MB.
> The following table shows the memory overhead of DCP and ModalGlue with seqlen=32k, 64k, and 128k (256k omitted due to limit). All memory in MB.
>
> - 32k
>
> | Memory | Causal | Lambda | Blockwise | Shared Question |
> |:-----------------------:|:------:|:------:|:---------:|:---------------:|
> |        DCP (base)       |   40.0 |   40.0 |      40.0 |            40.0 |
> |        DCP (peak)       |  474.1 |  488.4 |     446.8 |           461.7 |
> |     ModalGlue (base)    |   40.7 |   40.7 |      43.2 |            40.7 |
> |     ModalGlue (peak)    |  191.0 |  191.0 |     183.8 |           191.0 |
>
> - 64k
>
> | Memory | Causal | Lambda | Blockwise | Shared Question |
> |:-----------------------:|:------:|:------:|:---------:|:---------------:|
> |        DCP (base)       |   80.0 |   80.0 |      80.0 |            80.0 |
> |        DCP (peak)       |  943.8 |  958.7 |     893.5 |           911.2 |
> |     ModalGlue (base)    |   82.0 |   82.0 |      82.0 |            82.0 |
> |     ModalGlue (peak)    |  511.5 |  511.5 |     511.5 |           511.5 |
>
> - 128k
>
> | Memory | Causal | Lambda | Blockwise | Shared Question |
> |:-----------------------:|:------:|:------:|:---------:|:---------------:|
> |        DCP (base)       |  160.0 |  160.0 |     160.0 |           160.0 |
> |        DCP (peak)       | 1876.9 | 1910.8 |    1789.8 |          1823.0 |
> |     ModalGlue (base)    |  166.2 |  166.2 |     166.2 |           166.2 |
> |     ModalGlue (peak)    | 1341.2 | 1341.2 |    1341.2 |          1341.2 |
>
> Base memory consumption difference is because of the bitfield attention mask, because DCP frees the attention mask after compiling to generate blocks and schedule. Note that the attention mask is shared across multiple layers, thus memory overhead will be amortized. More importantly, in all cases, ModalGlue reports lower peak memory consumption than DCP.
>
> ## Using Real Datasets
>
> We run ModalGlue with real vision language dataset, HuggingFace’s FineVision. FineVision has several subdatasets, and here we run two out of them (LLaVA-150k and CocoQA). We run 10 iterations and report the execution time.
>
> |                          | 10 Iteration Times (ms)                                                             |
> |--------------------------|--------------------------------------------------------------------------------|
> | LLaVA 150k w/o ModalGlue | 4158.3, 4362.4, 4169.6, 4180.1, 4413.2, 4319.5, 4462.8, 4438.2, 4264.6, 4530.0 |
> | LLaVA 150k w/ ModalGlue  | 3353.8, 3496.8, 3492.4, 3554.2, 3497.1, 3575.9, 3670.0, 3548.8, 3417.6, 3614.0 |
> | CocoQA w/o ModalGlue     | 3992.8, 4022.2, 3980.7, 3941.5, 3966.7, 3885.1, 4031.9, 3824.4, 3970.3, 3994.9 |
> | CocoQA w/ ModalGlue      | 3114.7, 3163.3, 3099.8, 3089.6, 3081.9, 3038.8, 3218.3, 3241.7, 3151.6, 3089.5 |

---

> > ### Author Rebuttal · Reviewer_EnZh · 2026-04-03
> >
> > Thank you for your response. I will keep my positive score.

---

### Official Review · Reviewer_TDDT · 2026-03-10

**Soundness:** 2
**Presentation:** 3
**Significance:** 3
**Originality:** 2
**Overall Recommendation:** 4
**Confidence:** 4

**Summary:**

This paper presents ModalGlue, a distributed training framework for MLLMs that targets two practically important yet underexplored challenges. The first is that existing systems typically ignore the frozen status of multimodal modules when constructing pipeline parallelism schedules, which may result in imbalanced forward/backward workloads across stages. To address this, the authors propose a frozen status-aware pipeline parallelism method that explicitly incorporates module freezing into stage partitioning. The second challenge is that recent interleaved MLLMs often adopt non-causal attention patterns, especially bidirectional masks for vision tokens, which are not well supported by existing context parallel training methods. For this setting, the paper further proposes a workload-balanced context parallelism approach for non-causal attention masks. Overall, the paper is well-motivated, and the experimental results show the potential of ModalGlue in improving MLLM training efficiency.

**Compliance With Llm Reviewing Policy:**

Affirmed.

**Final Justification:**

The authors provided detailed experiments and comparison in rebuttal. Most of my concerns are resolved. I've raise my score to reflect it. Looking forward to see these additional results in the final version of the paper!

**Key Questions For Authors:**

Please see the weaknesses W1-W6.

**Limitations:**

Authors may discuss the limitations of their work.

**Strengths And Weaknesses:**

# Strengths
S1: [Soundness] The paper is clearly motivated and focuses on two challenges in MLLM training: pipeline imbalance caused by frozen multimodal modules and context-parallel workload imbalance under non-causal attention. The proposed methods are generally well aligned with these problem formulations. In particular, the paper considers the effect of frozen layers on forward/backward computation for PP, and addresses both inter-GPU and intra-GPU imbalance for CP in a reasonably self-consistent way.

S2: [Presentation] The paper is generally well organized and easy to follow. The background, limitations of prior work, method design, and implementation are presented in a clear and structured manner. The two components, frozen-aware PP and workload-balanced CP, are separated and explained clearly.

S3: [Significance] The problem is important. As MLLM architectures become more heterogeneous and multimodal sequences more complex, efficient distributed training becomes increasingly critical. The paper identifies two relevant gaps in existing systems and proposes corresponding solutions with clear empirical improvements.

S4: [Originality] The novelty is more in system integration than in fundamentally new techniques, but the unified treatment of two MLLM-specific challenges is still meaningful. In particular, the joint consideration of inter-GPU and intra-GPU balance in CP is a notable aspect.

# Weaknesses

W1: [Soundness] The baselines are relatively weak. The paper mainly compares against FSDP2 and Megatron, but does not include stronger prior systems or methods specifically designed for MLLM training, even though such works are discussed in the introduction. For example, Section 2 mentions prior efforts on dynamic non-causal context parallelism, such as DCP, but these are not included in the empirical comparison. Such methods should be considered as more relevant baselines.

W2: [Soundness] The experimental setting is narrow. The current setup freezes both the modality encoder and the LLM, while training only the projector. This configuration may be favorable to the proposed method, but it remains unclear whether the approach generalizes to more common partial-freezing settings, such as: (1) freezing only the modality encoder while training the LLM and projector, or (2) freezing only the LLM while training the modality encoder and projector. Since the paper aims to validate frozen-aware partitioning, it should evaluate a broader range of freezing patterns.

W3: [Soundness] The modeling and partitioning method for frozen-aware PP in Section 3.1 appears overly simplistic. Although the method accounts for the reduced backward cost of frozen modules, it does not truly optimize the 1F1B pipeline schedule under frozen-aware settings. The paper balances stages based on the total forward+backward computation time of modules, but under 1F1B scheduling, throughput depends not only on total F+B time, but also on the forward/backward ratio, stage position, and warmup/drain dependencies. Therefore, balancing by F+B alone is unlikely to yield schedule-optimal partitioning. In this sense, the paper identifies an interesting problem, but does not fully solve the frozen-aware PP scheduling issue it raises. It would also help to include a 1F1B scheduling illustration figure under frozen settings to make the motivation and method clearer.

W4: [Presentation / Significance] The discussion of non-causal attention masks in MLLMs should be more thorough. In practice, MLLM attention patterns are not uniform: some models use purely causal decoder-only attention, some allow bidirectional attention within image tokens on top of a causal backbone, and others introduce cross-attention modules. The mask shown in Figure 2(b) seems to assume that image tokens attend bidirectionally within the current image but not to preceding text, which may not represent all realistic designs. The paper should provide more background, cite more representative MLLM model architectures, and better justify whether non-causal masks are common and necessary in practice. It should also clarify whether the proposed method applies to arbitrary non-causal masks or only to specific structured cases.

W5: [Soundness] Section 5.4 states that the multimodal masks used in the experiments are simulated, but it is unclear whether these simulated masks accurately reflect the distribution found in real multimodal datasets. The paper should explain how these masks are generated and discuss how representative they are.

W6: [Soundness] The paper does not sufficiently discuss the additional system overhead introduced by workload-balanced CP. The method appears to require dynamic token remapping or token reorganization, which may incur nontrivial runtime cost. The paper should quantify this overhead, clarify whether it occurs at every training step, and analyze when the extra complexity is justified by the performance gain.

---

> ### Author Rebuttal · Authors · 2026-03-30
>
> We sincerely thank the reviewer for constructive feedback. We provide clarifications and additional results addressing each concern.
>
> ## More Various Setups with LoRA
>
> ModalGlue supports LoRA MLLM finetuning. Depending on the combination, forward/backward computation of each module in vision language model varies drastically as follows (time in ms):
> |            | Frozen  | LoRA    | Full    |
> |------------|---------|---------|---------|
> | Vision Fwd | 474.86  | 645.13  | 474.66  |
> | Vision Bwd | 0.43    | 2078.73 | 1781.81 |
> | LLM Fwd    | 470.04  | 782.31  | 470.16  |
> | LLM Bwd    | 1021.99 | 1760.61 | 1496.61 |
>
> Based on the profiled result, the following table shows performance difference:
> | Vision      | Frozen | Frozen | Frozen | LoRA   | LoRA   | LoRA   | Full   | Full   | Full   |
> |-------------|--------|--------|--------|--------|--------|--------|--------|--------|--------|
> | **LLM**         | **Frozen** | **LoRA**   | **Full**   | **Frozen** | **LoRA**   | **Full**   | **Frozen** | **LoRA**   | **Full**   |
> | Megatron-LM | 6404.6 | 6154.2 | 6969.1 | 8207.9 | 7430.0 | 8802.5 | 8247.4 | 7236.8 | 7025.6 |
> | ModalGlue   | 5004.4 | 5471.7 | 5307.0 | 6549.5 | 7199.9 | 8533.0 | 6513.0 | 6343.4 | 6761.1 |
>
>
> ## CP System Overhead (vs DCP)
>
> We compare ModalGlue against DCP on a single attention layer with 64 heads, 8 kv heads, and head_dim=128. For attention patterns, we implement 4 different attention patterns introduced in DCP. In this comment, we specifically focus on system overhead (1. preparation time and 2. memory overhead) than execution time. In short, ModalGlue outperforms DCP in both area.
>
> The following table shows the preparation time (DCP: graph generation and solving, ModalGlue: bitfield attention mask generation and token remapping). We show causal attention mask due to limited space.
>
> -mask: causal
>
> | Preparation | 32k  | 64k  | 128k  | 256k  |
> |-----------------------|------|------|-------|-------|
> | DCP                   | 24391| 27876| 33872| 44824 |
> | ModalGlue             | 2.10 | 3.80 | 12.83 | 39.32 |
>
> The following table shows the memory overhead of DCP and ModalGlue with seqlen=32k, 64k, 128k, and 256k. All in MB.
>
> - 32k
>
> | Memory | Causal | Lambda | Blockwise | Shared Question |
> |:-----------------------:|:------:|:------:|:---------:|:---------------:|
> |        DCP (base)       |   40.0 |   40.0 |      40.0 |            40.0 |
> |        DCP (peak)       |  474.1 |  488.4 |     446.8 |           461.7 |
> |     ModalGlue (base)    |   40.7 |   40.7 |      43.2 |            40.7 |
> |     ModalGlue (peak)    |  191.0 |  191.0 |     183.8 |           191.0 |
>
> - 64k
>
> | Memory | Causal | Lambda | Blockwise | Shared Question |
> |:-----------------------:|:------:|:------:|:---------:|:---------------:|
> |        DCP (base)       |   80.0 |   80.0 |      80.0 |            80.0 |
> |        DCP (peak)       |  943.8 |  958.7 |     893.5 |           911.2 |
> |     ModalGlue (base)    |   82.0 |   82.0 |      82.0 |            82.0 |
> |     ModalGlue (peak)    |  511.5 |  511.5 |     511.5 |           511.5 |
>
> - 128k
>
> | Memory | Causal | Lambda | Blockwise | Shared Question |
> |:-----------------------:|:------:|:------:|:---------:|:---------------:|
> |        DCP (base)       |  160.0 |  160.0 |     160.0 |           160.0 |
> |        DCP (peak)       | 1876.9 | 1910.8 |    1789.8 |          1823.0 |
> |     ModalGlue (base)    |  166.2 |  166.2 |     166.2 |           166.2 |
> |     ModalGlue (peak)    | 1341.2 | 1341.2 |    1341.2 |          1341.2 |
>
> - 256k
>
> | Memory | Causal | Lambda | Blockwise | Shared Question |
> |:-----------------------:|:------:|:------:|:---------:|:---------------:|
> |        DCP (base)       |  320.0 |  320.0 |     320.0 |           320.0 |
> |        DCP (peak)       | 3749.9 | 3808.9 |    3572.2 |          3643.2 |
> |     ModalGlue (base)    |  341.0 |  341.0 |     341.0 |           341.0 |
> |     ModalGlue (peak)    | 3155.0 | 3155.0 |    3155.0 |          3155.0 |
>
> Base memory consumption difference is because of the bitfield attention mask, because DCP frees the attention mask after compiling to generate blocks and schedule. Note that the attention mask is shared across multiple layers, thus memory overhead will be amortized. More importantly, in all cases, ModalGlue reports lower peak memory consumption than DCP.
>
> ## Frozen-Aware PP Overly Simplistic
>
> Focusing on 1F+1B is a hueristic but effective methodology to optimize pipeline parallelism overall. While it is true that throughput is not just decided only with F+B times, but considering the critical path of the pipeline parallelism, reducing the slowest F+B dominates overall pipeline parallel optimization. We refer to Oobleck Figure 5 and equation 4 to justify why this hueristic focusing on slowest F+B is effective [1]. Nevertheless, we will add a 1F1B scheduling under frozen setting to make it clear.
>
> 1. Jang, Insu, et al. "Oobleck: Resilient distributed training of large models using pipeline templates." SOSP 23

---

> > ### Author Rebuttal · Reviewer_TDDT · 2026-04-03
> >
> > The rebuttal is helpful and partially addresses my concerns, especially by adding results on more diverse freezing/LoRA/full settings and clarifying the CP overhead. However, several issues remain unresolved.
> > First, stronger and more relevant baselines are still missing. The added DCP comparison only reports preparation time and memory overhead on a single attention layer, rather than end-to-end training results.
> > Second, the realism of the simulated non-causal masks is still unclear, since the paper does not sufficiently explain how they are generated or how representative they are of practical MLLM workloads.
> > Third, the frozen-aware PP method still seems to be a heuristic based on balancing forward+backward cost, rather than a formulation that fully captures 1F1B scheduling effects.
> >
> >
> > Update: Glad to see the detailed results and comparison. Thank you for the detailed rebuttal! I will raise my score to reflect it. Looking forward to see these additional results in the final version of paper!

---

> > > ### Author Response · Authors · 2026-04-05
> > >
> > > Thank you for your comment! We apologize for not addressing those concerns in our initial rebuttal. Please let us address the remaining concerns further as follows.
> > >
> > > # 1. End-to-end training results vs DCP
> > > Please note that the main reason of not adding e2e training throughput is that DCP is not for multimodal. Therefore, e2e training throughput comparison should be done with just LLM and workload-balanced CP, excluding frozen aware PP.
> > > We compare workload-aware CP and DCP on Llama3-1b with the four types of attention masks that DCP used. Execution times includes planning + 1F + 1B (two seqlens due to limit).
> > >
> > > - 32k
> > > | Time (s)  | Causal | Lambda | Causal blockwise | Shared Question |
> > > |-----------|--------|--------|------------------|-----------------|
> > > | ModalGlue | 5.69   | 5.77   | 10.52            | 5.7             |
> > > | DCP       | 26.45  | 48.98  | 41.8             | 46.5            |
> > >
> > > - 128k
> > > | Time (s)  | Causal | Lambda | Causal blockwise | Shared Question |
> > > |-----------|--------|--------|------------------|-----------------|
> > > | ModalGlue | 37.2   | 37.4   | 29.0             | 33.9            |
> > > | DCP       | 46.5   | 107.1  | 100.8            | 102.4           |
> > >
> > > # 2. Realism of non-causal masks
> > > We apologize for not giving detailed explanations about how we form non-causal attention masks in our initial rebuttal response. Please find which non-causal masks we used in our evaluation. We will also make sure to clarify them in our revision.
> > >
> > > Our evaluation is an average of four different types of attention mask.
> > > 1. Early VLMs [1, 2] simply prefix all vision encoder tokens at the beginning of the sequence, forming MLLM attention mask as causal with prefix [3].
> > > 2. Later, MLLMs evolved to put modality tokens in the middle, the location of which are specified by special tokens, e.g., \<image\> or \<audio\>, like Figure 2 (b) [4, 5, 6].
> > > 3. One sample may have multiple images in different locations or multiple samples can be packed, forming more complicated patterns combining 1 and 2.
> > >
> > > We ran 16k, 32k, and 64k sequence length (results are also in Appendix C), and for each sequence length, we fill 20\%, 40\%, 60\% of the tokens with multimodal tokens. The ratio and location are randomly picked from normal distribution. The results shown in the paper are an average of attention masks, each of which ran 50 times.
> > >
> > >
> > > # 3. Effectiveness of hueristic based frozen-aware PP
> > > There appear to be misunderstanding regarding hueristic from our end. The reason we mentioned using 1F+1B is heuristic is that we do not separate forward and backward in deriving pipeline parallel scheduling, but consider 1F+1B as a single value.
> > > We still need an algorithm to partition model to pipeline stages, e.g., DC [3] or DP [4]. Any partitioning algorithm can be used, but the core idea of calculating B with frozen-status remains important. We will clarify it to Section 4 in the revision.
> > >
> > > The formulation that captures the entire 1F1B iteration time $T$, introduced in [7] (Figure 5), with $s$ stages and $K$ number of microbatches, where $k^\*$ indicates the index of the slowest stage ($0 \le k^\* < s$), can be formulated:
> > >
> > > $T = T1 + T2 + T3$
> > >
> > > where
> > >
> > > - $T1 = \sum_{i=0}^{s-1} (F_i + B_i)$
> > > - $T2 = (K - s + k^\* - 1)(F_{k^\*} + B_{k^\*})$
> > > - $T3 = \sum_{i=k^\*}^{s-1} (F_i + B_i)$
> > >
> > > Therefore, $T$ can be derived just with a set of F+B, without separated F and B. Also, given that $T \le (S+K-1)(F_{k^\*}+F_{k^\*})$ (as $F_i + B_i \le F_{k^\*} + B_{k^\*}$ for all $i$), minimizing $F_{k^\*} + B_{k^\*}$ bounds the iteration time $T$.
> > >
> > > We run DP from [8] using F+B as the layer cost to partition the model. For ModalGlue, we use Eq 1 for backward time, while full backward pass time is used for the frozen-unaware baseline. We provide a result of Qwen2Vision + Llama3-1b to 8 stages on 8 GPUs (V and L: frozen, projector: train), with #mb=32.
> > >
> > > -Partitioning results
> > >
> > > | Modality/# layers | S0  | S1  | S2  | S3   | S4  | S5  | S6   | S7  |
> > > |-------------------|-----|-----|-----|------|-----|-----|------|-----|
> > > | ModalGlue         | V/8 | V/8 | V/8 | V/10 | L/6 | L/5 | L/5  | L/2 |
> > > | Frozen Unaware    | V/6 | V/5 | V/6 | V/5  | V/6 | V/7 | L/11 | L/7 |
> > >
> > > - Per stage profile and iteration time
> > >
> > > | F/B (ms)       | S0       | S1       | S2       | S3       | S4        | S5        | S6         | S7         | Iter time (ms) |
> > > |----------------|----------|----------|----------|----------|-----------|-----------|------------|------------|----------------|
> > > | ModalGlue      | 67.1/0.0 | 64.5/0.0 | 64.5/0.0 | 65.5/0.0 | 21.2/45.9 | 19.7/45.5 | 19.7/45.5  | 24.1/32.0  | 5517.2         |
> > > | Frozen Unaware | 42.9/0.0 | 40.3/0.0 | 48.4/0.0 | 40.3/0.0 | 48.3/0.0  | 41.3/0.2  | 40.9/115.9 | 43.8/100.4 | 8323.5 (1.50x) |
> > >
> > >
> > > 1. LLaVA, NeurIPS 23
> > > 2. Paligemma, arXiv:2407.07726
> > > 3. Exploring the limits of transfer learning with a unified text-to-text transformer, JMLR 20
> > > 4. LLaVA 1.6, CVPR 24
> > > 5. Qwen2-vl arXiv:2409.12191
> > > 6. Gemma 3, arXiv:2503.19786
> > > 7. Oobleck, SOSP 23
> > > 8. Alpa, OSDI 22

---

### Official Review · Reviewer_8knw · 2026-03-11

**Soundness:** 2
**Presentation:** 3
**Significance:** 2
**Originality:** 2
**Overall Recommendation:** 3
**Confidence:** 3

**Summary:**

The paper introduces ModalGlue, an efficient distributed MLLM training framework that contemplates MLLM’s unique characteristics to optimize the training efficiency. ModalGlue improves the throughput of MLLM training by perceiving parameter freezing and enhancing the load balancing in the context parallelism. The framework is evaluated on 24 NVIDIA A40, achieving 2.26× higher end-to-end training throughput on average for MLLM training .

**Compliance With Llm Reviewing Policy:**

Affirmed.

**Final Justification:**

The authors have provided more detailed experimental data in the rebuttal, which strengthen their claims.

**Key Questions For Authors:**

See the weaknesses. If all the concerns are addressed, I am willing to raise my score.

**Limitations:**

yes

**Strengths And Weaknesses:**

### Strengths
1. The paper is well-structured and easy to follow. The presentation is clear.
2. Significant improvement. ModalGlue outperforms the baselines 2.26× on average (3.36× vs FSDP and 1.62× vs Megatron*).

### Weaknesses
1. Poor baseline comparison. FSDP and Megatron are both general training framework. I suggest to compare with at least one SOTA MLLM training framework. Also, there are many paper discussing the load-balance problem in context parallelism [1] [2].

2. No related work section.  I haven't seen any related work discussing any relevant solutions. Isn't this a very important part to evaluate the originality in the academic paper?


[1] Skrull: Towards Efficient Long Context Fine-tuning through Dynamic Data Scheduling

[2] ElasticMM: Efficient Multimodal LLMs Serving with Elastic Multimodal Parallelism

---

> ### Author Rebuttal · Authors · 2026-03-30
>
> We sincerely thank the reviewer for constructive feedback. We provide clarifications and additional results addressing each concern. Due to limited space, all results are small vision encoder + medium LLM on 4 nodes with 4*A40 each. Projector is always trainable. All time is in ms.
>
> ## Comparison to SOTA MLLM Frameworks
>
> As we state ModalGlue proposes higher-order optimization, SOTA MLLM frameworks are interoperable, not comparable. We apply ModalGlue to SOTA MLLM training to prove it and provide further performance improvement.
>
> However, some frameworks the reviewer mentioned are **not related** to pipeline parallelism and non-causal context parallelism that ModalGlue is optimizing, or not comparable with ModalGlue. The non-relevant frameworks include:
>
> - **OrchMLLM**: adjusting workload between data parallel ranks without other parallelism considered
> - **Skrull**: this is not even about multimodal LLM. Using causal attention mask and not about pipeline parallelism. Balancing causal attention context parallelism is already well-studied as mentioned in Section 2.2.
> - **ElasticMM**: focusing on orchestration of multiple inference requests with different stages, i.e, encode, prefill, and decode, not parallelization within each stage. Moreover, it only considers DP and TP in elastic parallelization, which has no relation to ModalGlue’s contribution.
>
> We compare 1F1B, DistTrain, and PipeWeaver w/ and w/o ModalGlue. We compare against DCP separately, since it focuses on context parallelism only.
>
> |            | w/o ModalGlue | w/ ModalGlue |
> |------------|---------------|--------------|
> | 1F1B       | 6404.6        | 5004.4       |
> | DistTrain  | 6135.7        | 4695.4       |
> | PipeWeaver | 5304.9        | 5161.6       |
>
> PipeWeaver adopts its own pipeline scheduling, so is not compatible with frozen-aware pipeline parallelism. Improvement only from context parallelism.
>
> ## Comparison to DCP
>
> We compare ModalGlue against DCP on a single attention layer with 64 heads, 8 kv heads, and head_dim=128. For attention patterns, we implement 4 different attention patterns introduced in DCP.
> In short, when sequence length is small, ModalGlue is slow due to slow inter-node network on our cluster setup. However, as sequence length is getting higher, computation effectively hides communication overhead and ModalGlue outperforms DCP.
>
> ### 4 nodes
>
> - mask: causal
>
> | Attention | 32k    | 64k    | 128k   | 256k    |
> |---------------------|--------|--------|--------|---------|
> | DCP                 | 54.63 |  149.18 | 503.56 | 1861.39 |
> | ModalGlue           | 211.51 | 158.15 | 380.88 | 1189.57 |
> - mask: lambda
> | Attention | 32k    | 64k    | 128k   | 256k    |
> |---------------------|--------|--------|--------|---------|
> | DCP                 | 46.36 |  124.72 | 417.07 | 1545.83 |
> | ModalGlue           | 177.41 | 143.56 | 375.49 | 1193.32 |
>
> - mask: causal blockwise
>
> | Attention | 32k    | 64k    | 128k   | 256k    |
> |---------------------|--------|--------|--------|---------|
> | DCP                 | 41.13 |  104.87 | 316.93 | 1083.57 |
> | ModalGlue           | 228.36 | 124.72 | 333.91 | 1002.70 |
>
> - mask: shared question
>
> | Attention | 32k    | 64k    | 128k   | 256k    |
> |---------------------|--------|--------|--------|---------|
> | DCP                 | 39.79 |  120.74 | 374.99 | 1358.43 |
> | ModalGlue           | 199.99 | 179.59 | 351.41 | 1190.41 |
>
> Note that DCP prep time (graph generation and solving) is unacceptably slow:
>
> - mask: causal
>
> | Preparation (ms) | 32k  | 64k  | 128k  | 256k  |
> |-----------------------|------|------|-------|-------|
> | DCP                   | 24391| 27876| 33872| 44824 |
> | ModalGlue             | 2.10 | 3.80 | 12.83 | 39.32 |
>
> - mask: lambda
>
> | Preparation (ms) | 32k  | 64k  | 128k  | 256k  |
> |-----------------------|------|------|-------|-------|
> | DCP                   | 45812 | 61837 | 92131 | 152419 |
> | ModalGlue             | 2.13 | 4.33 | 16.21 | 45.01 |
>
> - mask: causal blockwise
>
> | Preparation (ms) | 32k  | 64k  | 128k  | 256k  |
> |-----------------------|------|------|-------|-------|
> | DCP                   | 40428 | 56515 | 90586 | 157046 |
> | ModalGlue             | 2.29 | 4.36 | 14.14 | 45.64 |
>
> - mask: shared question
>
> | Preparation (ms) | 32k  | 64k  | 128k  | 256k  |
> |-----------------------|------|------|-------|-------|
> | DCP                   | 45709 | 60085 | 89161 | 152022 |
> | ModalGlue             | 2.09 | 4.22 | 13.26 | 40.67 |
>
>
> ## Lack of Related Work Section
>
> Due to high redundancy and limited number of pages, we put all relevant works into Section 2 (Background and Motivation). We provide 4D parallelism works, MLLM distributed training works, and balancing causal context parallel works. Except for DCP mentioned in the last paragraph of Section 2.2, there is no work that is directly related to parallelizing models with various frozen status or parallelizing non-causal context parallelism. However, we will add Related Work section in the final version.

---

> > ### Author Rebuttal · Reviewer_8knw · 2026-04-02
> >
> > Dear authors,
> >
> > Thank you for your reply and the new results.
> >
> > To clarify, I just want to know how your load balancing is different from existing methods for document masks.
> >
> > At the system level, I think there is no real difference between LLM and MLLM, or between causal and non-causal masks. The core problem is always the same: how to balance the uneven work in context parallelism under different attention mask.
> >
> > For example, methods like MagiAttention can already handle any attention mask.  I am very willing to raise my score if I find a novelty or distinction in this aspect.
> >
> > Thank you.

---

> > > ### Author Response · Authors · 2026-04-02
> > >
> > > > there is no real difference between LLM and MLLM, or between causal and non-causal masks. The core problem is how to balance the uneven work in context parallelism
> > >
> > > Thank you for pointing it out. While we strongly agree the core problem is how to balance the uneven work, we would like to state that there is a huge gap between balancing causal and non-causal masks. As introduced in Section 2.2, balancing causal mask is perfectly guaranteed even with **static** heuristic, however, balancing non-causal mask is not simple and requires more careful **dynamic** workload distribution, which is what ModalGlue, DCP, and MagiAttention try to achieve in their own way.
> > >
> > > After reviewing MagiAttention design, their idea is to decompose the whole attention mask to smaller AttnSlice, where each slice can be represented to either full mask or causal mask ([Figure 15](https://arxiv.org/pdf/2505.13211#page=25)). In terms of "representing document masks" specifically, ModalGlue BitfieldAttention and MagiAttention have equivalent capability.
> > >
> > > The novelty of ModalGlue BitfieldAttention lies in practicality in real world deployment and low assignment overhead.
> > > If context parallelism is used together, MagiAttention works more similarly to DCP; when we partition AttnSlices into chunks for CP, each chunk is equivalent to a DCP block. DCP stated that assigning blocks is a NP-complete, and MagiAttention says it is NP-hard. For this reason, both derive greedy hueristic algorithm; however, as we listed in our rebuttal, even with the hueristic algorithm the DCP assignment takes really long time, while ModalGlue overhead is an order of magnitude lower. Considering each sample requires its unique attention mask, high assignment overhead is unacceptable and leads to high end-to-end training throughput degradation, while attention performance itself looks better.
> > >
> > > Please note that DCP has end-to-end evaluation section in the paper, however, it is not clear whether such planning time is included, and we doubt since their reported iteration time is only 1~2 seconds. They also have planning time analysis, but when we run the microbenchmark the planning time is hugely impacted by the number of GPUs and sequence length, not just block size. Therefore, it is not clear how the planning actually affects end-to-end training performance. MagiAttention does not mention end-to-end performance.
> > >
> > > We unfortunately were not able to verify it in larger-scale cluster, however, the impact of overhead is already clear even in smaller cluster and will only be higher proportionally to the cluster size.

---

### Official Review · Reviewer_ZJVx · 2026-03-13

**Soundness:** 3
**Presentation:** 3
**Significance:** 3
**Originality:** 3
**Overall Recommendation:** 5
**Confidence:** 3

**Summary:**

This paper presents ModalGlue, a distributed training framework for MLLMs. The main idea is that multimodal models introduce extra heterogeneity that standard LLM parallelism does not handle well, especially when some components are frozen and when multimodal attention creates non-causal, uneven context-parallel workloads. To address this, the paper proposes frozen-status-aware pipeline parallelism and a workload-balanced context parallelism scheme that balances work at both the inter-GPU and intra-GPU level. Empirically, the paper reports 2.26× average throughput improvement over prior distributed training approaches, with gains across different model structures, modalities, and sizes.

**Compliance With Llm Reviewing Policy:**

Affirmed.

**Key Questions For Authors:**

1. How much of the gain depends on the specific Lava-style trainin here? Right now the experiments freeze the merged modality encoder and the LLM, and only train the projector modules. Can you show results for more common finetuning settings, such as LoRA on the LLM or partial encoder finetuning on the llm part?

2. How does ModalGlue compare to the most relevant recent MLLM training systems that are newer than FSDP and a Megatron-based extension? e.g., any of DistMM / DistTrain / Optimus / PipeWeaver / DCP?

3. How well do the gains scale with larger clusters or newer hardware?

**Limitations:**

The experiments are on synthetic data, a 24×A40 setup, and a fairly specific training regime where the encoder and LLM are frozen and only projector modules are trained, so I think the paper should be more explicit about that limited validation scope.

I would also like a clearer limitations paragraph on what is still unvalidated, especially whether the same methods preserve convergence and final model quality in real end-to-end MLLM training. Right now the evaluation is mostly throughput-focused, so that gap should be acknowledged directly.

Overall, I would not punish the paper heavily for this, but I do think the limitations / impact section should be expanded to better match the paper’s claims.

**Strengths And Weaknesses:**

Strengths

* The paper identifies two bottlenecks that are actually specific to MLLM training. The frozen/trainable split changes what good pipeline partitioning looks like, and multimodal non-causal attention creates a real load-balancing problem for context parallelism.

* The context-parallel part is also a good contribution. The paper goes beyond token-count balancing and points out that you can still get poor utilization because of uneven work distribution within a GPU, which is a nice systems insight.

* The reported gains are not marginal: the paper shows substantial throughput improvement overall, and both main components seem to contribute meaningfully.


Weaknesses

* The targeted training setup is relatively constrained. The experiments are on synthetic data, on a 24×A40 setup, and in a regime where the encoder and LLM are frozen and only projector modules are trained, so the paper does not really establish that this is a broadly applicable MLLM training framework.

* It is still importatn to double check the numerical difference and end-to-end training performance with the new implementation to guarantee the implemnetations is still correct. It shows throughput and utilization gains, but there is no result showing that these changes preserve convergence or end-task performance in a real training run.

* The scalability evidence is still limited for a systems paper. Everything is on one cluster size and one hardware generation, and there is not much sense of how the method behaves as the setup gets larger or more modern.

---

> ### Author Rebuttal · Authors · 2026-03-30
>
> We sincerely thank the reviewer for constructive feedback. We provide clarifications and additional results addressing each concern. Due to limited space, all results are small vision encoder + medium LLM on 4 nodes with 4*A40 each. Projector is always trainable. All time is in ms.
>
> ## More Various Setups with LoRA
>
> ModalGlue supports LoRA MLLM finetuning. Depending on the combination, forward/backward computation of each module in vision language model varies drastically as follows:
>
> |            | Frozen  | LoRA    | Full    |
> |------------|---------|---------|---------|
> | Vision Fwd | 474.86  | 645.13  | 474.66  |
> | Vision Bwd | 0.43    | 2078.73 | 1781.81 |
> | LLM Fwd    | 470.04  | 782.31  | 470.16  |
> | LLM Bwd    | 1021.99 | 1760.61 | 1496.61 |
>
> Based on the profiled result, the following table shows performance difference:
>
> | Vision      | Frozen | Frozen | Frozen | LoRA   | LoRA   | LoRA   | Full   | Full   | Full   |
> |-------------|--------|--------|--------|--------|--------|--------|--------|--------|--------|
> | **LLM**         | **Frozen** | **LoRA**   | **Full**   | **Frozen** | **LoRA**   | **Full**   | **Frozen** | **LoRA**   | **Full**   |
> | Megatron-LM | 6404.6 | 6154.2 | 6969.1 | 8207.9 | 7430.0 | 8802.5 | 8247.4 | 7236.8 | 7025.6 |
> | ModalGlue   | 5004.4 | 5471.7 | 5307.0 | 6549.5 | 7199.9 | 8533.0 | 6513.0 | 6343.4 | 6761.1 |
>
> 4 combinations out of 9: lora+lora, lora+full, full+lora, and full+full, have trainable parameters on both encoder and LLM sides, thus the performance improvement of frozen-aware pipeline parallelism is reduced. However, please note that there are various training methods that trains only either encoder or LLM as follows. A few examples:
>
> - V frozen, L frozen: LLaVA 1.5 (stage 1) [1], MiniGPT-4 [2], BLIP-2 [3]
> - V frozen, L full: LLaVA 1.5 (stage 2), VILA (stage 1) [4]
> - V full, LLM frozen: Frozen [5]
>
> ## Comparison to SOTA MLLM Frameworks
>
> As we state ModalGlue proposes higher-order optimization, SOTA MLLM frameworks are interoperable, not comparable. We apply ModalGlue to SOTA MLLM training to prove it and provide further performance improvement.
> We compare 1F1B, DistTrain, and PipeWeaver w/ and w/o ModalGlue. We compare against DCP separately, since it focuses on context parallelism only.
>
> |            | w/o ModalGlue | w/ ModalGlue |
> |------------|---------------|--------------|
> | 1F1B       | 6404.6        | 5004.4       |
> | DistTrain  | 6135.7        | 4695.4       |
> | PipeWeaver | 5304.9        | 5161.6       |
>
> PipeWeaver adopts its own pipeline scheduling, so is not compatible with frozen-aware pipeline parallelism. Improvement only from context parallelism.
>
> ## Comparison to DCP
>
> We compare ModalGlue against DCP on a single attention layer with 64 heads, 8 kv heads, and head_dim=128. For attention patterns, we implement 4 different attention patterns introduced in DCP.
> In short, when sequence length is small, ModalGlue is slow due to slow inter-node network on our cluster setup. However, as sequence length is getting higher, computation effectively hides communication overhead and ModalGlue outperforms DCP.
>
> ### 4 nodes
>
> - mask: causal
>
> | Attention | 32k    | 64k    | 128k   | 256k    |
> |---------------------|--------|--------|--------|---------|
> | DCP                 | 54.63 |  149.18 | 503.56 | 1861.39 |
> | ModalGlue           | 211.51 | 158.15 | 380.88 | 1189.57 |
>
> - mask: lambda
>
> | Attention | 32k    | 64k    | 128k   | 256k    |
> |---------------------|--------|--------|--------|---------|
> | DCP                 | 46.36 |  124.72 | 417.07 | 1545.83 |
> | ModalGlue           | 177.41 | 143.56 | 375.49 | 1193.32 |
>
> - mask: causal blockwise
>
> | Attention | 32k    | 64k    | 128k   | 256k    |
> |---------------------|--------|--------|--------|---------|
> | DCP                 | 41.13 |  104.87 | 316.93 | 1083.57 |
> | ModalGlue           | 228.36 | 124.72 | 333.91 | 1002.70 |
>
> - mask: shared question
>
> | Attention | 32k    | 64k    | 128k   | 256k    |
> |---------------------|--------|--------|--------|---------|
> | DCP                 | 39.79 |  120.74 | 374.99 | 1358.43 |
> | ModalGlue           | 199.99 | 179.59 | 351.41 | 1190.41 |
>
> Note that DCP prep time (graph generation and solving) is unacceptably slow. An example of causal:
>
> | Preparation (ms) | 32k  | 64k  | 128k  | 256k  |
> |-----------------------|------|------|-------|-------|
> | DCP                   | 24391| 27876| 33872| 44824 |
> | ModalGlue             | 2.10 | 3.80 | 12.83 | 39.32 |
>
> 1. Liu, Haotian, et al. "Improved baselines with visual instruction tuning." CVPR 24
> 2. Zhu, Deyao, et al. "MiniGPT-4: Enhancing Vision-Language Understanding with Advanced Large Language Models." ICLR 24
> 3. Li, Junnan, et al. "Blip-2: Bootstrapping language-image pre-training with frozen image encoders and large language models." ICML 23
> 4. Lin, Ji, et al. "Vila: On pre-training for visual language models." CVPR 24
> 5. Tsimpoukelli, Maria, et al. "Multimodal few-shot learning with frozen language models." NeurIPS 21

---

> > ### Author Rebuttal · Reviewer_ZJVx · 2026-04-02
> >
> > My questions have been fully addressed. I'll keep my positive score. Nice work!

---

### Decision · Program_Chairs · 2026-04-30

**Decision:**

Accept (regular)

**Comment:**

This paper introduces ModalGlue, an efficient distributed MLLM training framework tailored to the unique characteristics of MLLMs in both model and data parallelization. It proposes frozen-aware pipeline parallelism and workload-balanced context parallelism, achieving a 2.26× average improvement in training throughput over prior methods.


This paper received three positive ratings: Accept (Reviewer ZJVx), Weak Accept (Reviewer TDDT), Weak Accept (Reviewer EnZh), and one negative rating: Weak Reject (Reviewer 8knw). After the rebuttal, most concerns raised by Reviewers ZJVx, TDDT, and EnZh were addressed. Reviewer 8knw’s main concern was the differnece from MagiAttention. In response, the authors clarified that the novelty of ModalGlue’s BitfieldAttention lies in its practicality for real-world deployment and its low assignment overhead. Although Reviewer 8knw did not provide further comments, the AC believes this concern was partially addressed.

Therefore, the final decision is accept.